# Comprehensive profiling of the gut microbiota in patients with chronic obstructive pulmonary disease of varying severity

Yu-Chi Chiu[1,2]☯, Shih-Wei Lee[1]☯, Chi-Wei Liu[1], Rebecca Chou-Jui Lin[1], Yung-Chia Huang[1], Tzuo-Yun Lan[2]*, Lawrence Shih-Hsin Wu[3]*

1 Department of Internal Medicine, Taoyuan General Hospital, Ministry of Health and Welfare, Taoyuan, Taiwan, 2 Institute of Hospital & Health Care Administration, National Yang Ming Chiao Tung University, Taipei, Taiwan, 3 Graduate Institute of Biomedical Sciences, China Medical University, Taichung, Taiwan

☯ These authors contributed equally to this work.
* lshwu@hotmail.com (LSW); tylan@nycu.edu.tw (TYL)

**Data Availability Statement:** All relevant data are within the manuscript and its Supporting information files.

## Abstract

Chronic obstructive pulmonary disease (COPD) is a chronic respiratory disease that reduces lung and respiratory function, with a high mortality rate. Severe and acute deterioration of COPD can easily lead to respiratory failure, resulting in personal, social, and medical burden. Recent studies have shown a high correlation between the gut microbiota and lung inflammation. In this study, we investigated the relationship between gut microbiota and COPD severity. A total of 60 COPD patients with varying severity according to GOLD guidelines were enrolled in this study. DNA was extracted from patients' stool and 16S rRNA data analysis conducted using high-throughput sequencing followed by bioinformatics analysis. The richness of the gut microbiota was not associated with COPD severity. The gut microbiome is more similar in stage 1 and 2 COPD than stage 3+4 COPD. *Fusobacterium* and *Aerococcus* were more abundant in stage 3+4 COPD. Ruminococcaceae NK4A214 group and *Lachnoclostridium* were less abundant in stage 2–4, and *Tyzzerella 4* and *Dialister* were less abundant in stage 1. However, the abundance of a *Bacteroides* was associated with blood eosinophils and lung function. This study suggests that no distinctive gut microbiota pattern is associated with the severity of COPD. The gut microbiome could affect COPD by gut inflammation shaping the host immune system.

## Introduction

Chronic obstructive pulmonary disease (COPD) is an inflammatory lung disease and characterized by progressive obstruction of airflow, resulting in symptoms such as shortness of breath, cough and increased sputum [1]. Exacerbation of COPD often results in high mortality and morbidity, rapid decline in lung function, and increased health care expense [2]. Though cigarette smoking is associated with COPD, not all smokers develop the disease [2]. Furthermore,

**Funding:** This work was supported by Taoyuan General Hospital, Ministry of Health and Welfare, Taoyuan, Taiwan (PTH10702). The funders had no role in study design, data collection and analysis, decision to publish, or preparation of the manuscript.

**Competing interests:** The authors have declared that no competing interests exist.

even though COPD can lead to exacerbations, not all patients are susceptible to the symptoms. Therefore, COPD is a heterogeneous disease that may be affected by many factors that are not fully understood.

The pathogenesis of COPD is thought to involve inflammatory mediators and bacterial or viral infections [3]. Especially, systemic inflammation [4] and airway inflammation [5] are often associated with exacerbation. Traditional culturing techniques have found evidence of bacterial and viral colonization in the airways of COPD patients with exacerbations [6, 7]. These pathogens persist in the respiratory tract, creating a diverse environment in the airways and lungs. Though their presence in relation to exacerbations is not clearly defined, it has been assumed that any pathogen exposure may induce surfactant abnormalities, hinder mucociliary clearance, and increase the patient's susceptibility to chronic inflammation, worsening respiratory symptoms and accelerating disease progression.

This gut dysbiosis in humans is related to inflammation of the gastrointestinal tract itself, but also in the airways, such as in asthma and COPD [8, 9]. Accumulating evidence has highlighted the influence of the gut microbiota on lung immunity, referred to as the gut–lung axis, though the underlying pathways and mechanisms are still areas of intensive research [10]. Recently, faecal microbiome of COPD patients and healthy controls were investigated and found several species with different distribution between two groups, including members of *Streptococcus* and the family Lachnospiraceae, also correlate with reduced lung function [11]. Despite the close association of gut microbiota with inflammation and many lung diseases, the association between the differences in gut microbiota profiles and the severity of COPD disease is still unknown.

Several studies have found significant differences in the distribution of respiratory microbiota between healthy individuals and COPD patients, and between different levels of COPD severity [12]. There are growing interest in the effect of probiotics on lung disorders, such as asthma and COPD [13], which should indicate whether the gut microbiome is associated with COPD exacerbation or severity. In light of the above information, we investigated the relationship between gut microbiota and COPD severity.

## Materials and methods

### Subjects

A total of 60 COPD patients (> 20 years old) with varying severity according to GOLD guidelines [14] were enrolled in this study. DNA was extracted from patients' stool. Patients with cancer or other immune-related diseases and viral infections (e.g., Hepatitis B, Hepatitis C, HIV, etc.) were excluded from this study.

The stool samples were obtained from patients with moderate COPD and patients with severe COPD in stable condition (at least 3 months without exacerbation or use of antibiotics for any other reason). Diagnosis and classification of COPD was established according to GOLD recommendations [14]. The patient groups were defined as A (stage 1), B (stage 2), and C (stage 3+4) according to the classification of airflow limitation in the severity of COPD [14]. DNA was extracted from the stool samples using Qiagen QIAamp DNA Stool Mini Kit (Qiagne, Hilden, Germany) and subjected to next-generation sequencing (NGS). DNA quality was verified before and after rRNA depletion treatment by the Agilent 2100 Bioanalyzer. The DNA samples were also treated with RNase. All DNA processing were performed under aseptic conditions.

The study protocol conformed to the ethical guidelines of the 1975 Declaration of Helsinki and was approved by the Ethics Committee of Taoyuan General Hospital, Taoyuan, Taiwan

(reference number: TYGH106037). Written informed consent was obtained from each patient enrolled in the study.

## MetaVx™ library preparation and illumina MiSeq sequencing

NGS library preparations and Illumina MiSeq sequencing were performed at GENEWIZ, Inc. (Suzhou, China). DNA samples were quantified using a Qubit 2.0 Fluorometer (Invitrogen, Carlsbad, CA, USA). A total of 30–50 ng of DNA was used to generate amplicons using a MetaVx™ Library Preparation kit (GENEWIZ, Inc., South Plainfield, NJ, USA).

V3 and V4 hypervariable regions of prokaryotic 16S rDNA were selected to generate amplicons and subsequent taxonomy analysis. GENEWIZ designed a panel of proprietary primers aimed at relatively conserved regions bordering the V3 and V4 hypervariable regions of bacteria and Archaea16S rDNA. The V3 and V4 regions were amplified using forward primers containing the sequence `CCTACGGRRBGCASCAGKVRVGAAT` and reverse primers containing the sequence `GGACTACNVGGGTWTCTAATCC`. First-round PCR products were used as templates for second-round amplicon enrichment PCR. At the same time, indexed adapters were added to the ends of the 16S rDNA amplicons to generate indexed libraries ready for downstream NGS on Illumina Miseq.

The DNA libraries were validated by an Agilent 2100 Bioanalyzer (Agilent Technologies, Palo Alto, CA, USA) and quantified using a Qubit 2.0 Fluorometer. DNA libraries were multiplexed and loaded on an Illumina MiSeq instrument according to the manufacturer's instructions (Illumina, San Diego, CA, USA). Sequencing was performed using a 2 x 300 paired-end (PE) configuration; image analysis and base calling were conducted by the MiSeq Control Software (MCS) embedded in the MiSeq instrument.

## Data analysis

The QIIME data analysis package was used for 16S rRNA data analysis. The forward and reverse reads were joined and assigned to samples based on barcode, and truncated by cutting off the barcode and primer sequence. Quality filtering of joined sequences was performed and sequences that did not fulfil the following criteria were discarded: sequence length < 200 bp, no ambiguous bases, mean quality score ≥ 20. The sequences were then compared to the reference database (RDP Gold database) using the UCHIME algorithm (https://drive5.com/uchime/uchime_download.html) to detect chimeric sequences, and the chimeric sequences removed. The effective sequences were used in the final analysis. Sequences were grouped into operational taxonomic units (OTUs) using the clustering program VSEARCH (1.9.6) [15] against the Silva 119 database pre-clustered at 97% sequence identity. The Ribosomal Database Program (RDP) classifier was used to assign a taxonomic category to all OTUs at a confidence threshold of 0.8. The RDP classifier uses the Silva 132 database, which has taxonomic categories predicted to the species level. Sequences were rarefied prior to calculation of alpha and beta diversity statistics. Alpha diversity indexes were calculated in QIIME (version 1.9.1) [16] from rarefied samples using the Shannon index for diversity and the Chao1 index for richness. For beta diversity analysis, Principal Component Analysis (PCA) was performed and plotted based on Brary-Curtis distance matrix by R version 3.1.1 (https://cran.r-project.org/bin/windows/base/old/3.1.1/). The pheatmap package (https://cran.r-project.org/src/contrib/Archive/pheatmap/) was used for ecological analysis and heatmaps. By using metastats, differential analysis of taxonomic composition at genus level between groups can be performed based on the differential abundance between different groups. Differences in the abundance of microbial communities in two groups can be evaluated using strict statistical methods. The multiple hypothesis test was performed and false discovery rate (FDR) of the rare frequency

data determined to evaluate the significance of the observed difference. The FDR-adjusted p-values were calculated using the Benjamini-Hochberg procedure.

Correlation analysis used statistical models to study the correlation between random variables and investigates whether a dependency exits between the phenomena and the nature and level of association. Spearman correlation coefficient was determined using R version 3.1.1 based on the OTU abundance and clinical features (blood eosinophil percentage and lung function). P values were also obtained. Heatmaps were generated to illustrate the relationship between clinical features and OTUs.

## Results

### Demographic and clinical features of study subjects

We totally enrolled 60 male COPD patients and 20 patients diagnosed as stage 1, 20 patients as stage 2, and 20 patients as stage 3+4. The demographic and clinical features of enrolled patients were listed in Table 1. The patients in mild COPD group were elder than patients in moderate and severe COPD groups. Clinical features (with statistical significance), such as pulmonary function test, reflected the different among this three patients groups. The medication also had significant difference among the three study groups.

The stool DNA samples were eluted in 200 μl AE buffer. The average of DNA concentration was 5.52 ng/μl (range 1.05–12.43 ng/μl).

### OTUs

According to the results of the OTU cluster analysis, the common and unique OTUs of different samples/groups were analysed and showed in a Venn diagram (S1 Fig). The statistics for the OUTs sequence number in each sample are given in S1 Table (for data sharing).

### Alpha and beta diversity analyses among three COPD groups

The Chao1 index and Shannon index were used to evaluate the alpha diversity of the microbiome in the three COPD groups. The results are given in Fig 1a and 1b. We did not find any significant difference in OTU richness and diversity among three groups. Using the PCA analysis and Brary-Curtis distance matrix for beta diversity analysis, we also found no significant different in comparison of bacterial communities among three COPD groups (Fig 1c).

### Taxonomic distribution among three COPD groups

At phylum level, we observed the phylum abundance of each groups, and found Bacteroidetes was more abundant in grade 1 than grade 2–4 COPD (Fig 2). The top 30 abundant taxa in each sample or group were clustered and plotted in a heatmap at the species and genus level (Figs 3 and 4). The samples of three groups did not form a distinct cluster according to the cluster of sample analysis in species and genus level (Figs 3a and 4a). In the cluster of groups analysis based on euclidean distance, the top 30 species and genus in groups A and B were more similar than those in group C (Figs 3b and 4b). At genus level (Fig 4b), *Fusobacterium* and *Aerococcus* were more abundant in group C (stage 3 and 4). The Ruminococcaceae NK4A214 group and *Lachnoclostridium* were less abundant in group B/C (stage 2–4), and *Tyzzerella* 4 and *Dialister* were less abundant in group A (stage 1).

### Differential abundance

The differential analysis was carried out at the genus level. The abundance distribution of the five genera with the largest between-group difference is shown in Fig 5. The X-axis indicates

**Table 1. The demographic and clinical characteristics of the study participants.**

| Variables | Mild COPD | Moderate COPD | Severe COPD | *p* value |
|---|---|---|---|---|
| **Age (years)** | | | | |
| Mean ± SD (range) | 78 ± 11 (51–95) | 72 ± 10 (51–91) | 68 ± 8 (50–81) | 0.008[a] |
| **BH (m)** | | | | |
| Mean ± SD | 1.64 ± 0.07 | 1.66 ± 0.07 | 1.63 ± 0.07 | 0.630[a] |
| **BW (Kg)** | | | | |
| Mean ± SD | 61.79 ± 9.75 | 63.86 ± 10.24 | 58.94 ± 8.88 | 0.277[a] |
| **BMI** | | | | |
| Mean ± SD | 22.85 ± 3.51 | 23.35 ± 3.71 | 22.09 ± 3.40 | 0.535[a] |
| **WBC (per ul)** | | | | |
| Median (range) | 6700 (4090–9060) | 6470 (4620–13740) | 9335 (2580–19590) | 0.081[b] |
| **Eosinophil (%)** | | | | |
| Median (range) | 2.10 (0.5–10.9) | 1.75 (0–8.4) | 1.60 (0–14.8) | 0.354[b] |
| **Eosinophil (per ul)** | | | | |
| Median (range) | 146 (29–701) | 118 (0–597) | 162 (0–881) | 0.216[b] |
| **IgE (kU/L)** | | | | |
| Median (range) | 53.45 (1.50–7758.30) | 20.80 (1.5–436.30) | 62.50 (6.80–2130.30) | 0.296[b] |
| **Smoking (n)** | | | | |
| Yes | 15 | 17 | 18 | 0.432[c] |
| No | 5 | 3 | 2 | |
| **CAT** | | | | |
| Mean ± SD | 6.60 ± 3.50 | 9.45 ± 6.92 | 14.25 ± 6.30 | <0.001[a] |
| Score < 10 (n) | 17 | 12 | 6 | 0.002[c] |
| Score≧10 (n) | 3 | 8 | 14 | |
| **mMRC** | | | | |
| Mean ± SD | 0.35 ± 0.59 | 1.05 ± 1.19 | 1.80 ± 1.06 | <0.001[a] |
| Score < 2 (n) | 19 | 12 | 8 | 0.001[c] |
| Score≧2 (n) | 1 | 8 | 12 | |
| **Pulmonary function test** | | | | |
| **Pre-bronchodilator** | | | | |
| **FVC (L)** | | | | |
| Mean ± SD | 3.18 ± 0.61 | 2.88 ± 0.65 | 2.48 ± 0.54 | 0.002[a] |
| **FVC (% predicted)** | | | | |
| Mean ± SD | 115.65 ± 24.85 | 96.45 ± 17.70 | 84.80 ± 21.21 | <0.001[a] |
| **FEV$_1$ (L)** | | | | |
| Mean ± SD | 2.05 ± 0.48 | 1.52 ± 0.39 | 0.97 ± 0.24 | <0.001[a] |
| **FEV$_1$ (% predicted)** | | | | |
| Mean ± SD | 98.40 ± 20.23 | 65.95 ± 13.30 | 42.15 ± 11.38 | <0.001[a] |
| **FEV$_1$/FVC ratio (%)** | | | | |
| Mean ± SD | 64.80 ± 11.38 | 54.05 ± 13.18 | 40.10 ± 10.85 | <0.001[a] |
| **Post-bronchodilator** | | | | |
| **FVC ± SD (L)** | | | | |
| Mean ± SD | 3.33 ± 0.51 | 3.03 ± 0.66 | 2.64 ± 0.64 | 0.003[a] |
| **FVC (% predicted)** | | | | |
| Mean ± SD | 120.75 ± 19.30 | 101.85 ± 18.23 | 90.05 ± 22.60 | <0.001[a] |
| **FEV$_1$ (L)** | | | | |
| Mean ± SD | 2.10 ± 0.43 | 1.63 ± 0.39 | 1.03 ± 0.27 | <0.001[a] |
| **FEV$_1$ (% predicted)** | | | | |

*(Continued)*

**Table 1.** (Continued)

| Variables | Mild COPD | Moderate COPD | Severe COPD | *p* value |
|---|---|---|---|---|
| Mean ± SD | 101.30 ± 18.41 | 71.20 ± 13.59 | 44.60 ± 12.48 | <0.001[a] |
| **FEV$_1$/FVC ratio (%)** | | | | |
| Mean ± SD | 63.15 ± 7.97 | 54.80 ± 11.44 | 40.10 ± 11.38 | <0.001[a] |
| **Medication** | | | | |
| LAMA (n) | 10 | 3 | 0 | 0.005[c] |
| LABA (n) | 1 | 1 | 0 | |
| LAMA+LABA (n) | 4 | 7 | 8 | |
| ICS+LABA (n) | 3 | 3 | 1 | |
| ICS+LAMA+LABA (n) | 2 | 6 | 11 | |

SD = standard deviation; COPD = chronic obstructive pulmonary disease; n = number of subjects; BH = body height; WB = body weight; BMI = body mass index; WBC = white blood cell; CAT = COPD Assessment Test; mMRC = Modified Medical Research Council; FVC = forced vital capacity; FEV$_1$ = Forced expiratory volume in one second; LAMA = long-acting muscarinic antagonist; LABA = long-acting beta agonist; ICS = inhaled corticosteroid.

[a]: The statistical analysis was tested by One-way ANOVA.

[b]: The statistical analysis was tested by Kruskal-Wallis test.

[c]: The statistical analysis was tested by $\chi^2$-test.

the names of the five genera and the Y-axis the relative abundance of each. We found four genera, including *Veillonella*, *Corynebacterium* 1, *Romboutsia*, and *Aerococcus*, that are more abundant in group C than groups A and B. *Megasphaera* was found at lower abundance in group A than groups B and C. The statistical significance may due to few outliers.

## OTU abundance correlated with blood eosinophil percentage and lung function

Correlation analysis revealed that some OTUs are associated with clinical features (Fig 6). OTU 19 (*Bacteroides* sp.) had a stronger negative correlation with eosinophil count

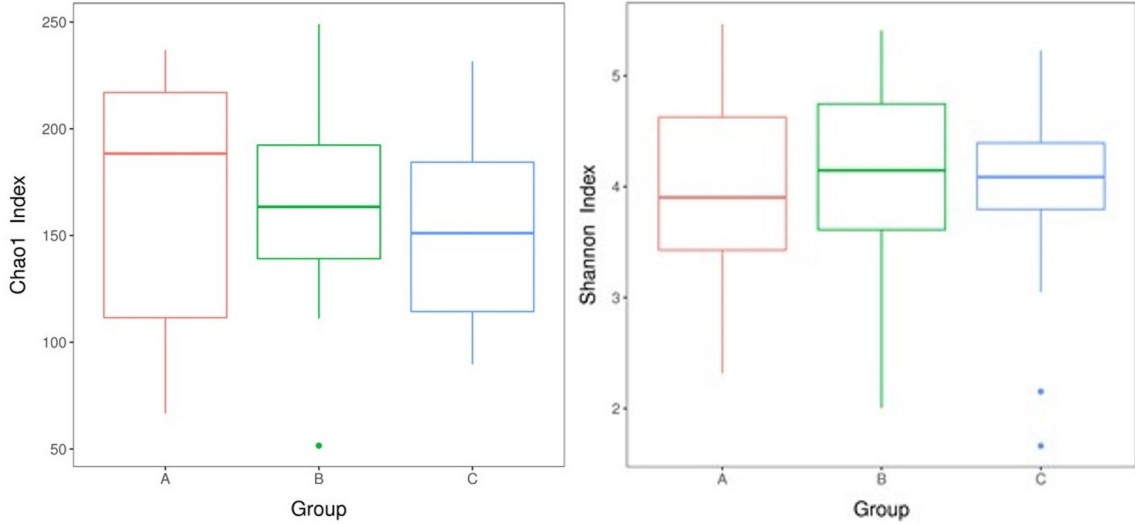

**Fig 1. The alpha and beta diversity analyses among three COPD groups.** a: Chao1 index boxplot of each group. The X-axis indicates the names of the groups and Y-axis the Chao 1 index. Each box diagram shows the minimum, first quartile, medium, third quartile, and maximum values of the Chao1 index of the corresponding sample; b: Shannon index boxplot of each group; c: PCA score plot. A: stage 1, B: stage 2, C: stage 3+4.

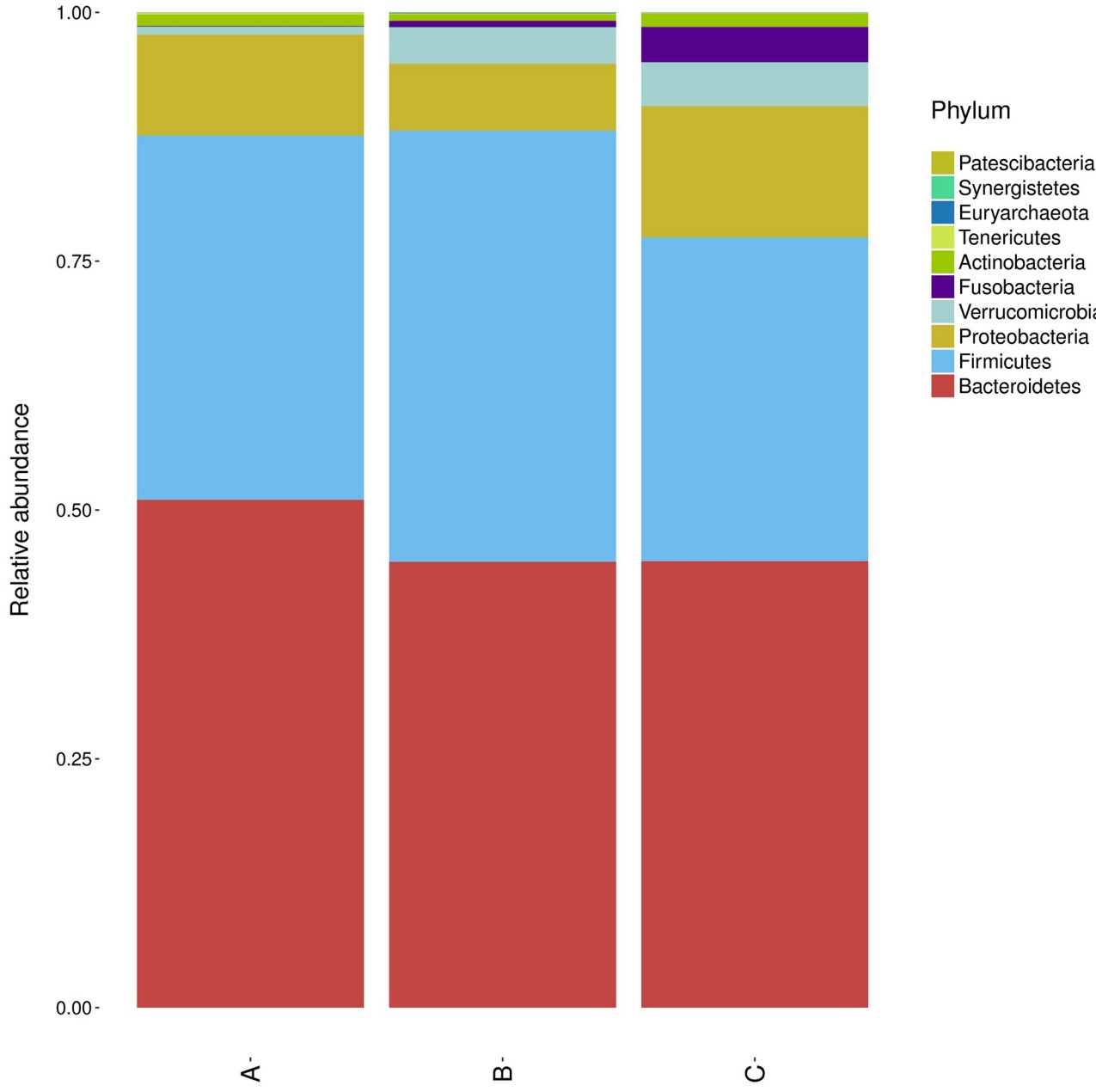

**Fig 2. Phylum abundance of each groups.** A: stage 1, B: stage 2, C: stage 3+4.

($P < 0.001$) and positively correlated with $FEV_1$ and FVC ($P < 0.05$). The statistical results of correlation analysis were shown in S2 Table.

## Discussion

COPD is becoming a leading cause of death and is increasingly prevalent worldwide [3–4]. The full spectrum of factors and mechanisms underlying the disease is still not completely understood. In this study, we investigated the gut microbiome in stool samples from 60 COPD patients with varying severity using 16S rRNA gene sequencing. In alpha and beta diversity

(a)

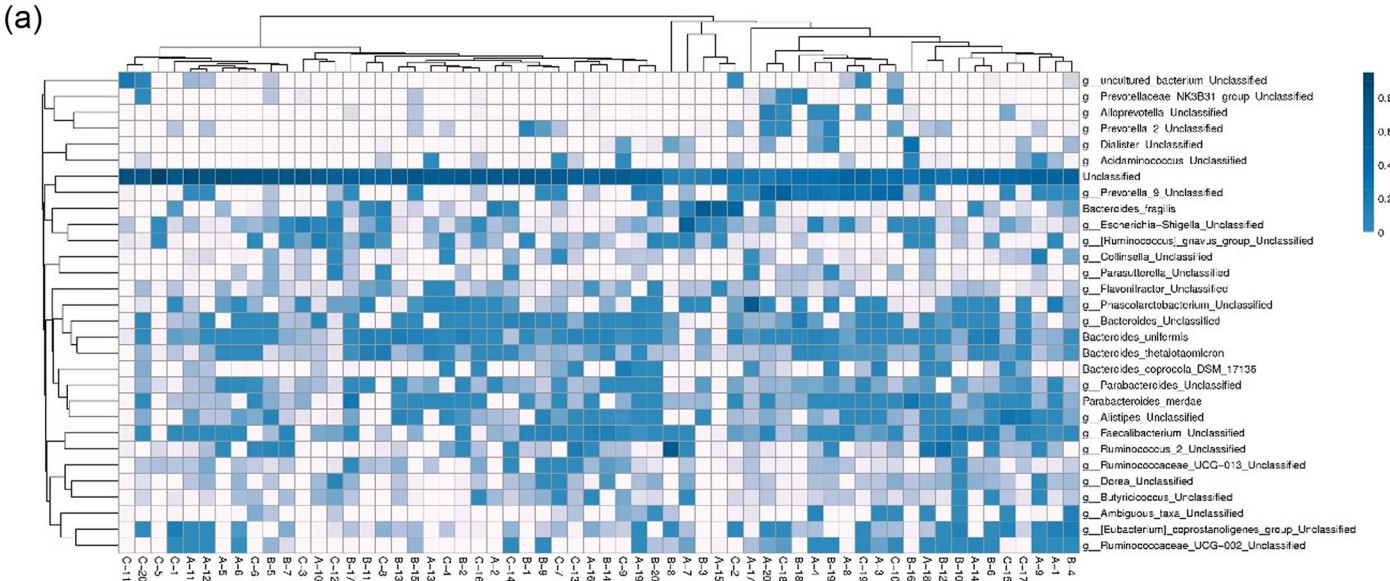

(b)

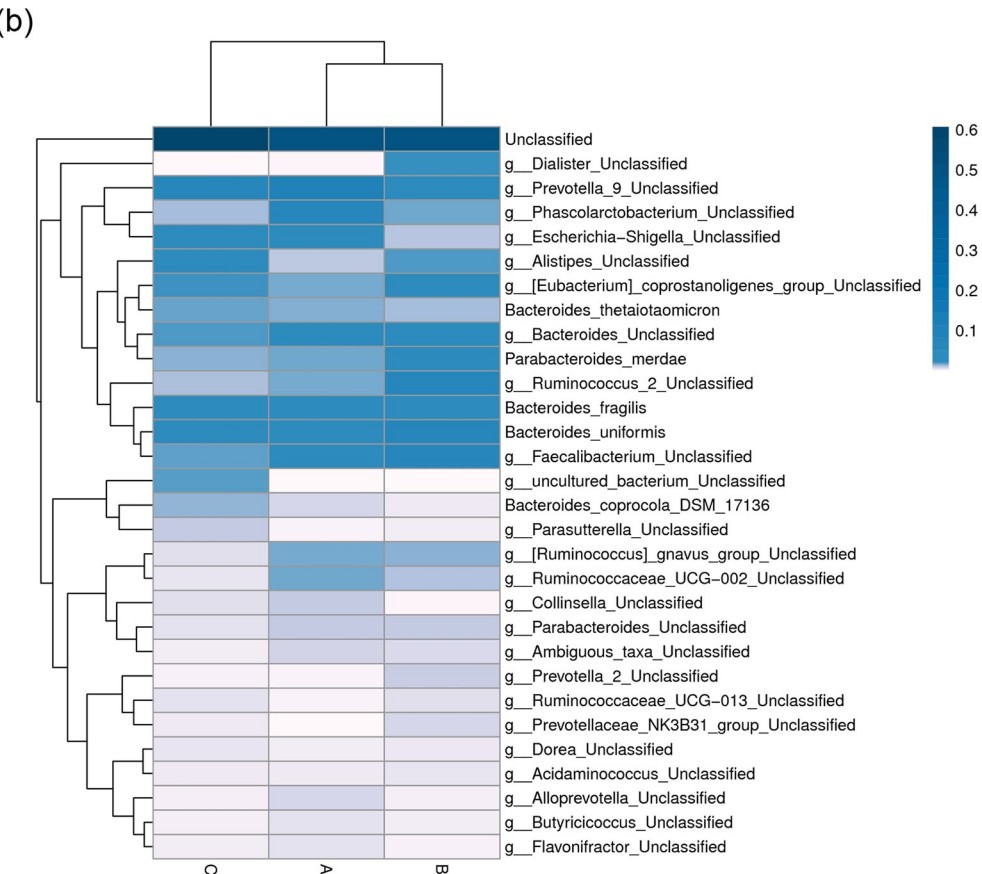

**Fig 3. The heatmap analysis of the top 30 species.** a: Cluster of samples. b: Cluster of groups. The columns represent samples and/or groups and the rows represent species. The dendrogram above the heatmap is the cluster result of the samples and/or groups and the dendrogram to the left is the species cluster. The colours in the heat map represent the relative abundance of the corresponding species in the corresponding sample or group.

(a)

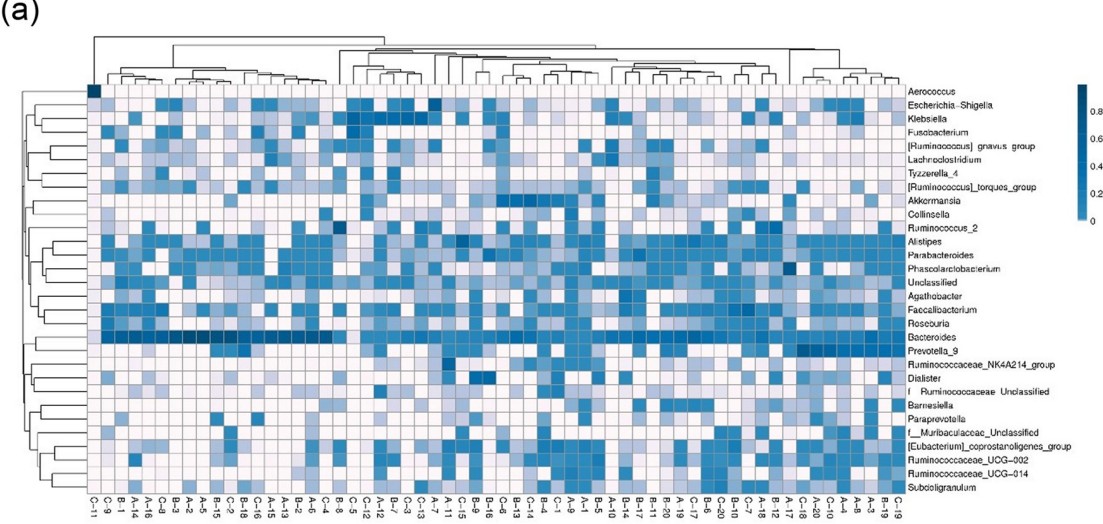

(b)

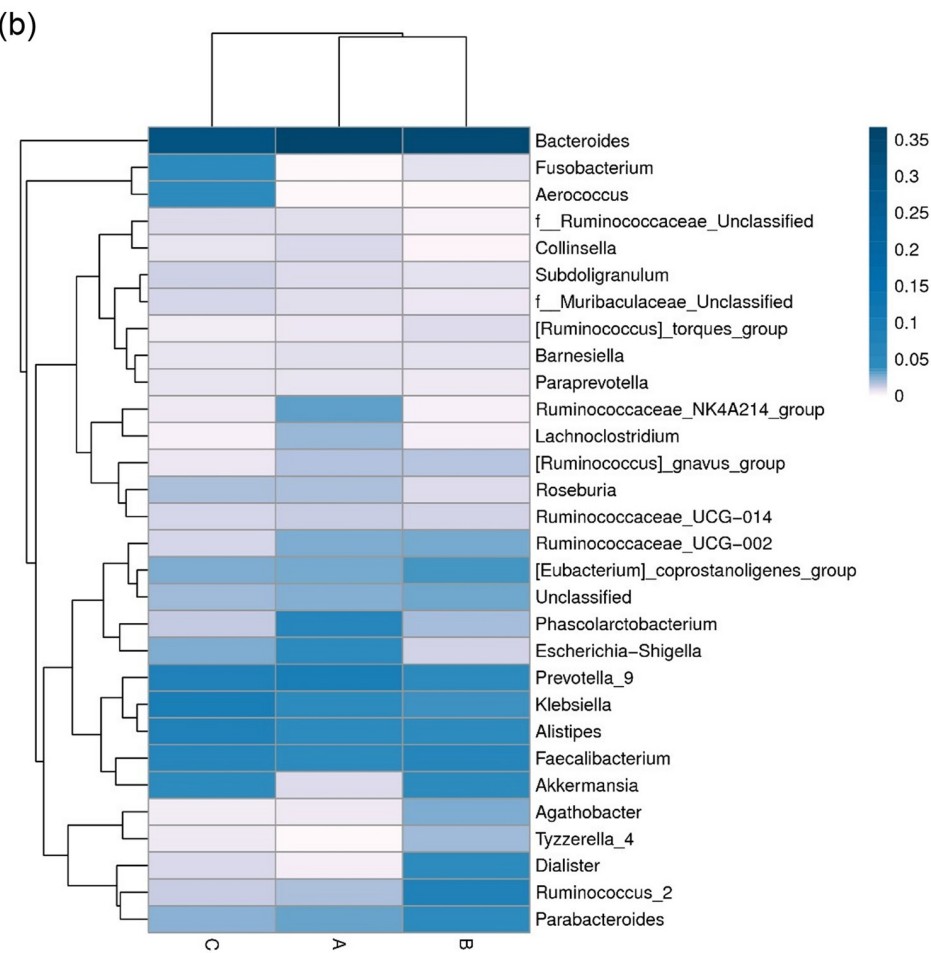

**Fig 4. The heatmap analysis of the top 30 genus.** a: Cluster of samples. b: Cluster of groups. The columns represent samples and/or groups and the rows represent genus. The dendrogram above the heatmap is the cluster result of the samples and/or groups and the dendrogram to the left is the genus cluster. The colours in the heat map represent the relative abundance of the corresponding genus in the corresponding sample or group.

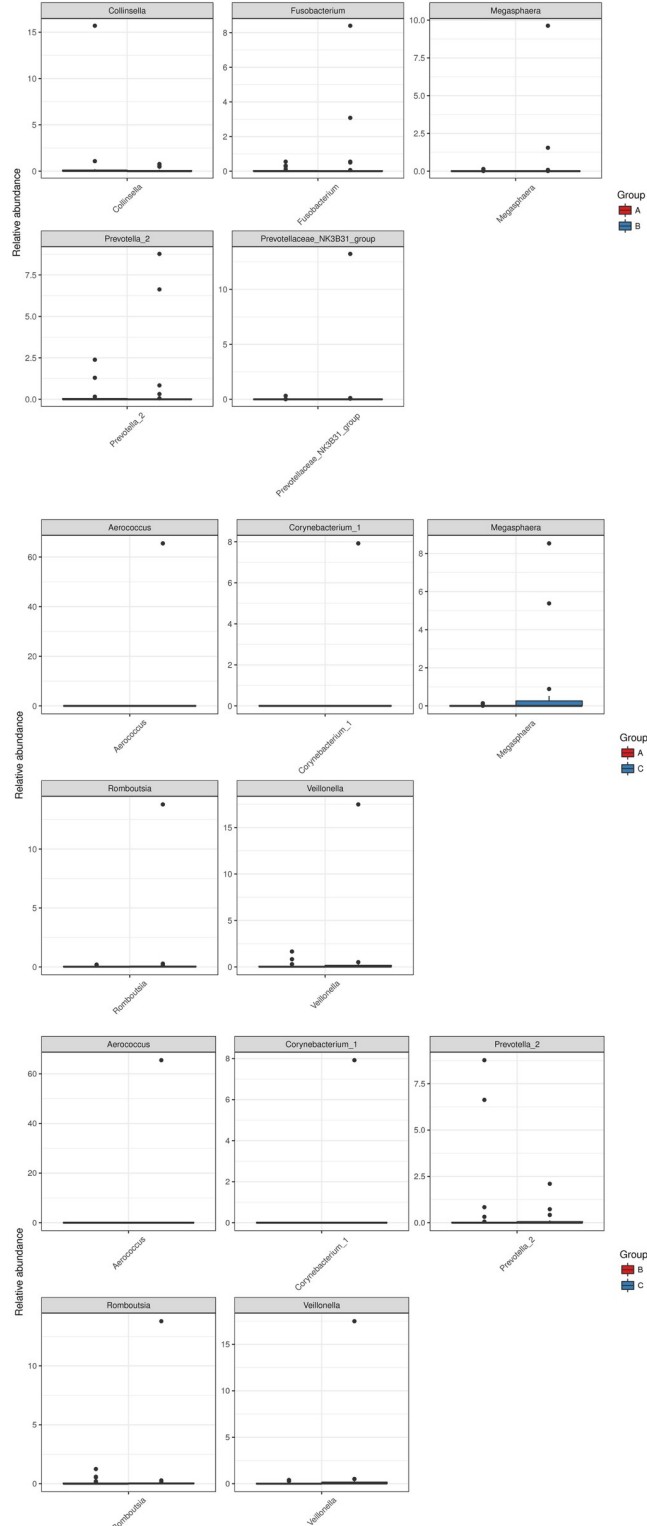

**Fig 5. Abundance distributions of the five genera with the largest between-group differences.** Top: group A vs. group B; middle: group A vs. group C; bottom: group B vs. group C.

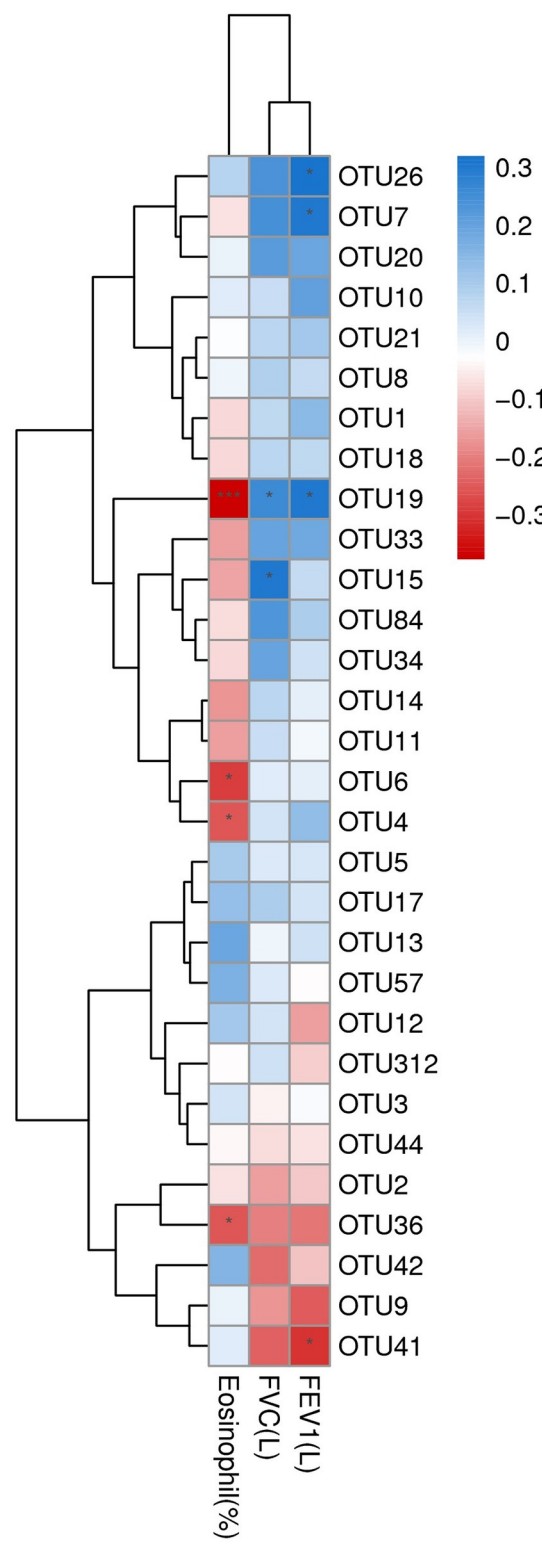

**Fig 6. The heatmap analysis of Spearman correlation between OTUs and blood eosinophil percentage and pulmonary function.** Spearman correlation coefficient (r) ranges between -1 and 1. r > 0 indicates positive correlation and r < 0 negative correlation. *p = 0.01–0.05, *** p<0.001. OTU 26: g__Ruminococcaceae_UCG-002, s__uncultured organism; OTU 7: g__Faecalibacterium, s__Ambiguous taxa; OTU 19: Bacteroides sp.; OTU 15: g__Bacteroides, s__unidentified; OTU 6: Bacteroides sp.; OTU 4: Parabacteroides_merdae; OTU 36: Bacteroides sp.; OTU 41: Fusobacterium sp. IDs of other non-significant OTUs are listed in S1 Table.

analyses, we did not find significant differences in bacterial richness and communities among stool samples of three COPD groups. In stage 3+4 COPD, the more abundant genera were *Fusobacterium* and *Aerococcus*. The Ruminococcaceae NK4A214 group and *Lachnoclostridium* in stage 2–4 COPD and *Tyzzerella* 4 and *Dialister* in stage 1 COPD were less abundant. Using Spearman correlation analysis, the abundance of a *Bacteroides* was associated with eosinophil count and lung function.

The gut microbiome has not been characterized previously in COPD patients. However, gut bacterial dysbiosis has been reported in response to cigarette smoke in both humans and mice. In gut microbiota composition, current smokers have more abundant of Bacteroidetes and less abundant of Firmicutes and Proteobacteria than never smokers [17]. In addition, healthy smokers had increased Bacteroides–Prevotella compare to non-smokers [18]. Significant alterations in microbiota composition have been reported in healthy smokers, which reverse upon smoking cessation, with marked increases in both overall microbial diversity and in the phyla Firmicutes and Actinobacteria, and a reduced proportion of Bacteroidetes and Proteobacteria compared to continuing smokers and non-smokers [19]. In previous mouse study, colonic19 bacterial dysbiosis was induced by chronic (24 weeks) exposure to cigarette smoke, with increased Lachnospiraceae sp. [20]. In our study, Bacteroidetes was more abundant in grade 1 COPD than grade 2–4 COPD (Fig 2). From the observation of Fig 5, the present of the abundance distribution of the five generagenera with the largest between-group difference may due to few outlier(s). That may indicate two issue: (1) the result is suspicious due to random sampling effect; (2) the outlier(s) observed only one or two study groups (not random sampling effect) suppose the bacteria associated with COPD severity only in part (not all) patients. However, the distinct pattern of gut microbiota defined by one or few bacteria is not revealed in this study. Furthermore, the severe COPD patients were higher ratio with inhaled corticosteroid (ICS) treatment that may indicate the medication did not alter gut microbiota obviously.

The blood eosinophil count was reported to associate with the risk of COPD exacerbation, mortality, decreased $FEV_1$, and response to corticosteroids [21]. The differential expression of the airway microbiome between eosinophilic and non-eosinophilic patients with COPD, during both stable disease [22] and acute disease exacerbation [23], suggest that dysregulation of this complex homoeostatic immunity is likely to feature in the pathogenesis of COPD. Bacterial counts for potentially pathogenic microorganisms negatively correlated with sputum eosinophil count, but not blood eosinophil count [24]. Our results indicate that the *Bacteroides* was associated with blood eosinophil percentage and lung function in COPD. This observation may indicate the different roles of gut and airway microbiomes in COPD *via* eosinophil inflammation. In previous mouse researches, the gut microbiome was essential to shaping the host immune system [25, 26]. The gut microbiome should influence the host immune system by modulating the blood eosinophil count, not directly affect COPD by pathogenic infection.

Our results shown that the more abundant genera in patients with severe COPD (stage 3 +4) were *Fusobacterium* and *Aerococcus*. *Fusobacterium nucleatum* is abundant in patients suffering from chronic gut inflammation, contributing to the pathogenesis of colorectal cancer [27]. *Aerococcus urinae* and *Aerococcus sanguinicola* are associated with urinary tract infections [28] but are unknown in the pathogenesis of gastrointestinal disease. We also found that the Ruminococcaceae NK4A214 group and *Lachnoclostridium* were less abundant, and *Tyzzerella 4* and *Dialister* more abundant in stage 2–4 COPD. Patients with non-alcoholic fatty liver disease (NAFLD) have lower abundance of Ruminococcaceae than those with non-NAFLD [29]. In patients with bipolar disorder, Ruminococcaceae is relatively decreased [30]. In children with autism spectrum disorders, the significant decrease in30 relative abundance of *Lachnoclostridium*, *Tyzzerella subgroup 4*, *Flavonifractor*, and unidentified *Lachnospiraceae* was

found [31]. The low frequency of *Dialister* observed was reported31 in non-inflamed ileal and colonic biopsy tissue from patients with spondyloarthritis and healthy controls [32]. Based on the aforementioned studies, we suggest that the different abundance of gut microbiome associated with varying COPD severity and involved gut inflammation in this study.

Some methodological factors limit the interpretation and inferences drawn from this study. First, the sample size was relatively small and caused to the results under power, which might be responsible, at least in part, for our findings with statistical significant was suspicious due to random sampling effect. Thus, a future study to further increase the number of each grouped subjects will help solidify our finding. Second, the heathy (or non-COPD) control did not enrolled for this study because the criteria of control are hard to define. However, the aim of this study is to identify the gut microbiota associated with COPD severity, not COPD *per se*. Third, we did not evaluate the extraction blank and collect the information about obesity and stool consistency of COPD patients in this study. Recent studies indicated the influence of reagent and laboratory contamination on sequence-based microbiome analysis [33] and the close associations of obesity and stool consistency with changes in gut microbiota [34–36]. The impact of these factors on microbiome analysis has been suggested, and it may affect the assay results.

Our results found no obviously relationship between gut microbiota and severity of COPD in humans. However, the association between blood eosinophils and gut microbiota in COPD patients was revealed in our study. Our results may provide useful information for developing new diagnostic or therapeutic methods to control COPD progression.

## Supporting information

**S1 Fig. OTU Venn diagram.**
(DOCX)

**S1 Table. The statistics for the OUTs sequence number in each sample.**
(XLS)

**S2 Table. Spearman correlation coefficient and p value of correlation analysis for OTUs and clinical features (blood eosinophil percentage and lung function).**
(XLS)

## Author Contributions

**Conceptualization:** Shih-Wei Lee, Tzuo-Yun Lan, Lawrence Shih-Hsin Wu.

**Formal analysis:** Lawrence Shih-Hsin Wu.

**Funding acquisition:** Yu-Chi Chiu, Shih-Wei Lee.

**Investigation:** Yu-Chi Chiu, Chi-Wei Liu, Rebecca Chou-Jui Lin, Yung-Chia Huang, Lawrence Shih-Hsin Wu.

**Methodology:** Shih-Wei Lee, Chi-Wei Liu, Rebecca Chou-Jui Lin, Yung-Chia Huang.

**Project administration:** Chi-Wei Liu, Lawrence Shih-Hsin Wu.

**Supervision:** Lawrence Shih-Hsin Wu.

**Writing – original draft:** Yu-Chi Chiu, Shih-Wei Lee, Tzuo-Yun Lan, Lawrence Shih-Hsin Wu.

**Writing – review & editing:** Tzuo-Yun Lan, Lawrence Shih-Hsin Wu.

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
