## [Decision Letter · Decision Letter 0]

4 Nov 2020

PONE-D-20-30223

Comprehensive profiling of the gut microbiota in patients with chronic obstructive pulmonary disease of varying severity

PLOS ONE

Dear Dr. Wu,

Thank you for submitting your manuscript to PLOS ONE. After careful consideration, we feel that it has merit but does not fully meet PLOS ONE’s publication criteria as it currently stands. Therefore, we invite you to submit a revised version of the manuscript that addresses the points raised during the review process.

Both reviewers saw value in your manuscript but raised concerns about methodological detail and data interpretation which should be addressed in your revision. As the study is relatively small, please ensure that conclusions are not over-stated.

We look forward to receiving your revised manuscript.

Kind regards,

Aran Singanayagam

Academic Editor

PLOS ONE

Journal Requirements:

2.We note that you are reporting an analysis of a microarray, next-generation sequencing, or deep sequencing data set. PLOS requires that authors comply with field-specific standards for preparation, recording, and deposition of data in repositories appropriate to their field. Please upload these data to a stable, public repository (such as ArrayExpress, Gene Expression Omnibus (GEO), DNA Data Bank of Japan (DDBJ), NCBI GenBank, NCBI Sequence Read Archive, or EMBL Nucleotide Sequence Database (ENA)). In your revised cover letter, please provide the relevant accession numbers that may be used to access these data. For a full list of recommended repositories, see http://journals.plos.org/plosone/s/data-availability#loc-omics or http://journals.plos.org/plosone/s/data-availability#loc-sequencing.

3.Thank you for stating the following in the Funding Section of your manuscript:

[This work was supported by Taoyuan General Hospital, Ministry of Health and

Welfare, Taoyuan, Taiwan (PTH10702).]

 [The funders had no role in study design, data collection and analysis, decision to publish, or preparation of the manuscript.]

4.Thank you for submitting the above manuscript to PLOS ONE. During our internal evaluation of the manuscript, we found significant text overlap between your submission and the following previously published works, some of which you are an author.

- https://journals.plos.org/plosone/article?id=10.1371%2Fjournal.pone.0159066

- https://onlinelibrary.wiley.com/doi/full/10.1038/cti.2017.6

- https://www.thelancet.com/journals/lanres/article/PIIS2213-2600(17)30217-5/fulltext

- https://www.sciencedirect.com/science/article/abs/pii/S0899900717301363?via%3Dihub

Please revise the manuscript to rephrase the duplicated text, cite your sources, and provide details as to how the current manuscript advances on previous work. Please note that further consideration is dependent on the submission of a manuscript that addresses these concerns about the overlap in text with published work.

Reviewers' comments:

Reviewer's Responses to Questions

**Comments to the Author**

1. Is the manuscript technically sound, and do the data support the conclusions?

Reviewer #1: Partly

Reviewer #2: Partly

2. Has the statistical analysis been performed appropriately and rigorously? 

Reviewer #1: Yes

Reviewer #2: No

3. Have the authors made all data underlying the findings in their manuscript fully available?

Reviewer #1: No

Reviewer #2: No

4. Is the manuscript presented in an intelligible fashion and written in standard English?

Reviewer #1: Yes

Reviewer #2: Yes

5. Review Comments to the Author

Reviewer #1: Wu and colleagues compare the composition of the gut microbiome across 60 patients with varying severity of COPD. Some thoughts:

1. I think the introduction would benefit from a line or two more making the case for examining the gut microbiome in patients with COPD. Although explained that SCFAs influence immune system and TMAO is associated with COPD mortality, these are a bit circumstantial – any other evidence to say why worth looking at this? Anything a bit more mechanistic? One potential idea – bile acids influence alveolar epithelial cells and lung fibroblasts: https://onlinelibrary.wiley.com/doi/full/10.1111/resp.12815 .

2. Materials and methods – overall OK, but there is a noticeable lack of referencing to basis of protocols used in this section. Which FDR correction method did you use? For the line ‘differential analysis of species composition…’, do you not mean ‘taxonomic’ rather than ‘species’, especially given that this is 16S data? Similarly - there is a mention made here of strain level analysis… it is very hard to believe that you were consistently able to analyse more than genus-level data confidently given that this is 16S data…? Have you got any references for the pipeline used if strain level analysis is being claimed? How did you compare taxonomic profiles between groups – R, STAMP, another method? A little detail on statistical testing is given, but this could be expanded. What code/ software did you use for your heatmaps?

3. Table 1 – you have said that you excluded patients using antibiotics within the past three months, but do you have any data on medication use here, as potentially of interest? e.g. antibiotic courses within the past year.

4. Figure 1 – text and figure do not make clear if only no signif difference in richness between groups, or also Shannon/ alpha diversity too? Perhaps update text and add bar to figure.

5. Figure 3 – along x axis, please put all A samples, then all B samples, then all C samples (as has been done for Figure 4 for genera) – very mixed up as it is and hard to interpret. I would remove unclassified, and try and clean up the labelling of the y axis (e.g. looks messy saying ‘g_Faecalibacterium_unclassified’ – suggest just change to ‘unclassified Faecalibacterium’).

6. Figure 5 – I don’t think any of these are actual strains, which does not surprise me; see my comment above. I am not sure this adds a huge amount, as is essentially more genus-level data.

7. Other clinical data – did you have enough follow-up to see what happened to these patients? Have you enough follow up data to try and correlate, e.g., number of future LRTIs since these samples taken with a particular taxonomic fingerprint?

8. Functional microbiome – your abstract mentions about the potential importance of SCFAs and TMA/TMAO to the gut-lung axis but doesn’t look at these at all in the analysis. While PLoS ONE is focused on technically sound work rather than novelty per se, this would substantially develop the paper. The ideal situation would be metabonomic analysis of stool, but appreciate this may not be feasible. What about using a bioinformatic tool to predict metagenomic content, e.g. Piphillin? Would be easy to do and could see if, e.g., predicted SCFA production was different between groups?

9. General – stated that study was approved by ethics committee of the hospital, but was there any other ethics panel review? Or a reference number? The raw microbiome data (e.g. .fastq files) should be available in the public domain (e.g. https://www.ebi.ac.uk/metagenomics/) unless the researchers have a very strong reason as to why this should not be the case.

Reviewer #2: General comments.

The authors have performed a 16S targeted amplicon sequencing analysis of gut microbiome composition in 60 patients stratified according to COPD disease severity. I was surprised to note that there are not yet any major studies conducted into this topic as the concept has been around for some time. I feel that greater numbers are required to fully establish a role for the gut microbiome in COPD. The study seems a little premature and somewhat rushed. Having said that, the authors are careful to acknowledge limitations and (for the most part) don’t make any overreaching conclusions.

Specific comments.

1. Please don’t use the term “strain” when referring to identified taxa. 16S does not provide credible strain-level resolution.

2. The authors state that participants had similar eating habits. The means by which this was established should be clarified or, if not specifically assessed ( e.g. food frequency questionnaire), the sentence should be removed.

3. The authors mention rRNA depletion in relation to sputum. I suspect the authors are thinking of another study here as neither sputum or transcriptomic analysis are presented in this manuscript.

4. The authors recruited 20 patients in each group for mild moderate and severe disease categories. I agree with the approach and it is good that they analyse the clinical parameters between groups but the sample size is too small for a gut microbiome study given significant inter-individual variability (c.f. PMID: 27126040, Zhernakova et al.). Nonetheless it could be informative if, perhaps, under powered.

5. “No significant difference in community richness.” Again, while it is challenging to assess power in microbiome studies, progress has been made and should be discussed (PMID: 27153704, Mattiello et al). The risk that type two error occurred here should be mentioned – remember the old adage; “no evidence of a difference is not evidence of no difference.” Base on obesity work one could speculate a difference should exist (obesity is a risk factor for COPD and the gut microbiome is altered in obese patients). I should say however, that a significant amount of variability in gut studies can be traced to stool consistency (i.e. Bristol stool chart, again see c.f. PMID: 27126040, Zhernakova et al.). This should be controlled for and leads to my obvious next question.

6. How was stool consistency assessed? If variable between severity groups it could represent a significant confounder. Was the Bristol stool chart used?

7. In relation to controls, blank extraction samples should be reported (both sequencing and extraction blanks). Although the authors report that DNA concentrations were quality controlled, and are likely to be microbe-rich given that these are stool samples, contamination is always a risk, especially when employing amplicon sequencing. Ideally, mock community analysis samples should also have been included.

7. “Group similarities at genus and species level”. In relation to the statement “In the cluster of sample analysis,group A and B were more similar than group C” - define what is meant by “more similar” in the text. What distance metric was employed.? I think A PCA analysis would work better here. Alternatively, if keeping the heat maps (3a 4a) colour could be used to indicate group membership of each patient.

8. Differential abundance. Did the authors use LEFSE or metastats? These are now standard methods. PERMANOVA could also be conducted. FDR anaysis is ok but won’t account for data sparsity. I commend the authors for their data availability (and the clarity of the data, which is easy to interpret and assess). However, I have performed Lefse analysis and this reveals no differences between groups A, B or C. I can reproduce the observation concerning Fusobacterium which does seem to be over represented in the severe group (group C).This does not completely invalidate the authors findings but serves to underscore that the sample size is small and different methods may give contrasting results based on the models employed. At least we can say there is some consistency as the Fusobacterium observation is reproduced. However, it is hard to know how reproducible or biologically important these findings are over all.

9. I don’t think species level analysis is advisable in the case of 16S analysis. I would cut at genus level as it is more reliable and gives a more realistic appraisal of the microbiome in my view.

10. Though the taxonomic data is accessible, the authors stop short of making the clinical metadata available. Without this I can’t reproduce the bacteriodes finding. I would recommend the authors implement Maaslin (https://huttenhower.sph.harvard.edu/maaslin/) to see if the observations regarding lung function and eosinophil counts hold up. If they do, this is potentially an interesting finding.

11. Final line of the discussion goes too far. “This should be useful information for developing new diagnostic or therapeutic markers to control COPD progression.” The findings must be validated in independent studies with larger patient numbers before we are anywhere near even considering diagnostics or therapeutics. This is a very preliminary glimpse of what MIGHT be going on in COPD that requires extensive additional validation work in future studies.

6. PLOS authors have the option to publish the peer review history of their article (what does this mean?). If published, this will include your full peer review and any attached files.

Reviewer #1: No

Reviewer #2: No

---

## [Author Response · Author response to Decision Letter 0]

2 Jan 2021

Responses to Reviewer #1:

We thank you for giving us a positive review. Because of your and Reviewer 2’s comments, we modified the manuscript.

Comment 1: I think the introduction would benefit from a line or two more making the case for examining the gut microbiome in patients with COPD. Although explained that SCFAs influence immune system and TMAO is associated with COPD mortality, these are a bit circumstantial – any other evidence to say why worth looking at this? Anything a bit more mechanistic? One potential idea – bile acids influence alveolar epithelial cells and lung fibroblasts: https://onlinelibrary.wiley.com/doi/full/10.1111/resp.12815.

Reply 1: Thank you for your valuable comments. Previous study indicated SCFAs can bind to G-protein coupled receptor 43 (GPR43) and strongly affect inflammatory responses [1,2]. TMAO induced the expression of pro-inflammatory genes and the production of inflammatory cytokines by activating NF-kappa B pathway [3,4]. We think the SCFAs and TMAO may over-emphasize and confuse the main focus in this manuscript. We deleted the statement about SCFAs/TMAO in introduction section. 

1. Brown AJ, Goldsworthy SM, Barnes AA, Eilert MM, Tcheang L, Daniels D, et al. The Orphan G protein-coupled receptors GPR41 and GPR43 are activated by propionate and other short chain carboxylic acids. J Biol Chem 2003; 278:11312-9

2. Morrison DJ, Preston T. Formation of short chain fatty acids by the gut microbiota and their impact on human metabolism. Gut Microbes 2016; 7: 189-200.

3. Seldin MM, Meng Y, Qi H, Zhu W, Wang Z, Hazen SL, et al. Trimethylamine N-Oxide Promotes Vascular Inflammation Through Signaling of Mitogen-Activated Protein Kinase and Nuclear Factor-κB. J Am Heart Assoc. 2016; 5: e002767.

4. Ma G, Pan B, Chen Y, Guo C, Zhao M, Zheng L, et al. Trimethylamine N-oxide in atherogenesis: impairing endothelial self-repair capacity and enhancing monocyte adhesion. Biosci Rep. 2017; 37: BSR20160244.

Comment 2: Materials and methods – overall OK, but there is a noticeable lack of referencing to basis of protocols used in this section. Which FDR correction method did you use? For the line ‘differential analysis of species composition…’, do you not mean ‘taxonomic’ rather than ‘species’, especially given that this is 16S data? Similarly - there is a mention made here of strain level analysis… it is very hard to believe that you were consistently able to analyse more than genus-level data confidently given that this is 16S data…? Have you got any references for the pipeline used if strain level analysis is being claimed? How did you compare taxonomic profiles between groups – R, STAMP, another method? A little detail on statistical testing is given, but this could be expanded. What code/ software did you use for your heatmaps?

Comment 2-1: Which FDR correction method did you use? 

Reply 2-1: To correct for multiple testing, we calculated the 'false discovery rate' (FDR)-adjusted p-values using the Benjamini-Hochberg procedure. We add the statement in the Materials and Methods (page 7) as you suggested.

Comment 2-2: For the line ‘differential analysis of species composition…’, do you not mean ‘taxonomic’ rather than ‘species’, especially given that this is 16S data?

Reply 2-2: Thank you for your valuable comments. We modified the text in the Materials and Methods (page 7) as you suggested.

Comment 2-3: Similarly - there is a mention made here of strain level analysis… it is very hard to believe that you were consistently able to analyse more than genus-level data confidently given that this is 16S data…?

Reply 2-3: We modified the text in the manuscript (page 2, 7, 10, 11, and 12) as you suggested

Comment 2-4: Have you got any references for the pipeline used if strain level analysis is being claimed?

Reply 2-4:

Thank you for comments, we has added some references for analysis methods: 

1. The sequences were then compared to the reference database (RDP Gold database) using the UCHIME algorithm (https://drive5.com/uchime/uchime_download.html)

2. VSEARCH (1.9.6): Torbjørn Rognes, ., Frédéric Mahé, ., Tomas Flouri, ., Daniel McDonald, ., Pat Schloss, ., & Ben J Woodcroft, . (2016, January 8). vsearch: VSEARCH 1.9.6 (Version v1.9.6). Zenodo. http://doi.org/10.5281/zenodo.44512

3. QIIME (version 1.9.1): Caporaso JG, Kuczynski J, Stombaugh J, Bittinger K, Bushman FD, Costello EK, Fierer N, Gonzalez Pena A, Goodrich JK, Gordon JI, Huttley GA, Kelley ST, Knights D, Koenig JE, Ley RE, Lozupone CA, McDonald D, Muegge BD, Pirrung M, Reeder J, Sevinsky JR, Turnbaugh PJ, Walters WA, Widmann J, Yatsunenko T, Zaneveld J, Knight R. 2010. QIIME allows analysis of high-throughput community sequencing data. Nature Methods 7(5): 335-336.

4. R version 3.1.1: (https://cran.r-project.org/bin/windows/base/old/3.1.1/).

Comment 2-5: How did you compare taxonomic profiles between groups – R, STAMP, another method? A little detail on statistical testing is given, but this could be expanded. Reply 2-5: In the differential analysis, we used metastats to compare the abundance distributions of the five genera with the largest between-group differences. We added the statement in Results (page 7) as you suggested.

Comment 2-6: What code/ software did you use for your heatmaps?

Reply 2-6: The R version 3.1.1 was used for heatmaps. We added the text in Results (page 7) as you suggested.

Comment 3: Table 1 – you have said that you excluded patients using antibiotics within the past three months, but do you have any data on medication use here, as potentially of interest? e.g. antibiotic courses within the past year.

Reply 3: In our study, we did not evaluate the antibiotic courses within the past year in 60 patients with COPD. But we found that three COPD patients in group A and group B and seven patients in group C used antibiotics within the past three months. Patients with severe COPD seem to take more antibiotics within the past three months than those with mild and moderate COPD.

Comment 4: Figure 1 – text and figure do not make clear if only no signif difference in richness between groups, or also Shannon/ alpha diversity too? Perhaps update text and add bar to figure.

Reply 4: In alpha diversity analysis, Shannon and Chao1 indexes were used respectively to evaluate OTU richness and diversity among three COPD groups. No significant difference in OTU richness and diversity among three COPD groups was found. We modified the text to the Materials and Methods (page 6) and the Results (page 8) as you suggested.

Comment 5: Figure 3 – along x axis, please put all A samples, then all B samples, then all C samples (as has been done for Figure 4 for genera) – very mixed up as it is and hard to interpret. I would remove unclassified, and try and clean up the labelling of the y axis (e.g. looks messy saying ‘g_Faecalibacterium_unclassified’ – suggest just change to ‘unclassified Faecalibacterium’).

Reply 5: Thank you for your valuable comments. The x-axis of heatmaps (order of samples and groups) were generated automatically by analysis. “g_Faecalibacterium_unclassified” means in genus level and unclassified in species level and also original output from software. We agree your comments but the post-production of the output is not easy for us. Sorry about that.

Comment 6: Figure 5 – I don’t think any of these are actual strains, which does not surprise me; see my comment above. I am not sure this adds a huge amount, as is essentially more genus-level data.

Reply 6: Thank you for your valuable comment. We modified the text in the results (pages 8-9) as you suggested. 

Comment 7: Other clinical data – did you have enough follow-up to see what happened to these patients? Have you enough follow up data to try and correlate, e.g., number of future LRTIs since these samples taken with a particular taxonomic fingerprint?

Reply 7: Thank you for your valuable comments. In this study, we did not evaluate the correlation between follow up clinical data and gut microbiota in 60 patients for this study, because it is difficult to long-term control the eating habits and antibiotics usage of 60 subjects in this study. These factors have been shown to change the distribution of gut microbiota and may affect the correlation between follow up clinical data and gut microbiota. However, the association between gut microbiota and follow up clinical data is worth investigating in the future.

Comment 8: Functional microbiome – your abstract mentions about the potential importance of SCFAs and TMA/TMAO to the gut-lung axis but doesn’t look at these at all in the analysis. While PLoS ONE is focused on technically sound work rather than novelty per se, this would substantially develop the paper. The ideal situation would be metabonomic analysis of stool, but appreciate this may not be feasible. What about using a bioinformatic tool to predict metagenomic content, e.g. Piphillin? Would be easy to do and could see if, e.g., predicted SCFA production was different between groups?

Reply 8: We stated the potential importance of SCFAs and TMA/TMAO to the gut-lung axis in introduction, not in abstract. We did not perform SCFA metabolomics analysis in this study. For avoiding the misleading to our study aims, the statements for SCFAs and TAMO have been deleted in introduction section. A new reference about a study for faecal microbiome of COPD patients and healthy controls has been added.

Comment 9: General – stated that study was approved by ethics committee of the hospital, but was there any other ethics panel review? Or a reference number? The raw microbiome data (e.g. .fastq files) should be available in the public domain (e.g. https://www.ebi.ac.uk/metagenomics/) unless the researchers have a very strong reason as to why this should not be the case.

Reply 9: No, this study was only approved by the Institutional Review Board/Ethics in Taoyuan General Hospital, Ministry of Health and Welfare. (reference number: TYGH106037). We had added a supplementary file contains the raw microbiome data as you suggested. The clinical data should be not provided in public domain owing to ethics issues, but can require from authors under reasonable request 

Responses to Reviewer #2:

We appreciate your kind comments. We revised the text to indicate the limits of this study. Our responses to your comments are as follows.

Comment 1: Please don’t use the term “strain” when referring to identified taxa. 16S does not provide credible strain-level resolution.

Reply 1: We modified the text in the manuscript (page 2, 7, 10, 11, and 12) as you suggested.

Comment 2: The authors state that participants had similar eating habits. The means by which this was established should be clarified or, if not specifically assessed (e.g. food frequency questionnaire), the sentence should be removed.

Reply 2: Thank you for your kind comment. We deleted the text in the Materials and Methods (pages 4) as you suggested.

Comment 3: The authors mention rRNA depletion in relation to sputum. I suspect the authors are thinking of another study here as neither sputum or transcriptomic analysis are presented in this manuscript.

Reply 3: Thank you for your kind comment. The association between microbiota in sputum and COPD disease have been investigated in our previous paper [Lee SW, Kuan CS, Wu LSH, Weng JTY. Metagenome and Metatranscriptome Profiling of Moderate and Severe COPD Sputum in Taiwanese Han Males. PLoS One 2016; 11: e0159066]. We modified the text in the Materials and Methods (pages 4) as you suggested.

Comment 4: The authors recruited 20 patients in each group for mild moderate and severe disease categories. I agree with the approach and it is good that they analyze the clinical parameters between groups but the sample size is too small for a gut microbiome study given significant inter-individual variability (c.f. PMID: 27126040, Zhernakova et al.). Nonetheless it could be informative if, perhaps, under powered.

Reply 4: Thank you for your kind comment. We agree this preliminary study is under power. Thus, a future study to further increase the number of each grouped subjects will help solidify our finding. We added the statement about the limitations in our study to the Discussion (page 12) as you suggested.

Comment 5: “No significant difference in community richness.” Again, while it is challenging to assess power in microbiome studies, progress has been made and should be discussed (PMID: 27153704, Mattiello et al). The risk that type two error occurred here should be mentioned – remember the old adage; “no evidence of a difference is not evidence of no difference.” Base on obesity work one could speculate a difference should exist (obesity is a risk factor for COPD and the gut microbiome is altered in obese patients). I should say however, that a significant amount of variability in gut studies can be traced to stool consistency (i.e. Bristol stool chart, again see c.f. PMID: 27126040, Zhernakova et al.). This should be controlled for and leads to my obvious next question.

Reply 5: Thank you for your valuable comments. However, we did not evaluate the relationship obesity and gut microbiota in this study. Previous study indicated the association between obesity and gut microbiota [34]. According to Department of Health in Taiwan, obesity was BMI ≧ 27 kg/m2. In our study, there were seven obese COPD patients (three in group A and B, one in group C). By using χ2 test, no significant difference in distribution of obese COPD patients was found in three COPD groups. Give these information, our observations in this study may not be influenced by obesity. We added the statement about the limitations in our study to the Discussion (page 12) as you suggested.

Comment 6: How was stool consistency assessed? If variable between severity groups it could represent a significant confounder. Was the Bristol stool chart used?

Reply 6: Thank you for your valuable comments. However, we did not use the Bristol stool chart to evaluate stool consistency. Recent studies indicated the close relationship between stool consistency and changes in gut microbiota [References 35-36]. We added the statement about the limitations in our study to the Discussion (page 12) as you suggested.

Comment 7: In relation to controls, blank extraction samples should be reported (both sequencing and extraction blanks). Although the authors report that DNA concentrations were quality controlled, and are likely to be microbe-rich given that these are stool samples, contamination is always a risk, especially when employing amplicon sequencing. Ideally, mock community analysis samples should also have been included.

Reply 7: Thank you for your valuable comments. In this study, all DNA processing were performed under aseptic conditions. But, we did not evaluate the extraction blank. Previous study indicated that reagent and laboratory contamination can influence sequence-based microbiome analysis [reference 33]. We add the statement about the limitations in our study to the Discussion (page 12) as you suggested. 

Comment 8: “Group similarities at genus and species level”. In relation to the statement “In the cluster of sample analysis, group A and B were more similar than group C” - define what is meant by “more similar” in the text. What distance metric was employed.? I think A PCA analysis would work better here. Alternatively, if keeping the heat maps (3a 4a) colour could be used to indicate group membership of each patient.

Reply 8: Thank you for your valuable comments. In the Group similarities at genus and species level of the Results, we indicated that the distribution of top 30 abundant taxa at species and genus level in groups A and B were more similar than those in group C. In Figure 3b and 4b, we used the euclidean distance to evaluate the similarities in three COPD group. 

Comment 9: Differential abundance. Did the authors use LEFSE or metastats? These are now standard methods. PERMANOVA could also be conducted. FDR anaysis is ok but won’t account for data sparsity. I commend the authors for their data availability (and the clarity of the data, which is easy to interpret and assess). However, I have performed Lefse analysis and this reveals no differences between groups A, B or C. I can reproduce the observation concerning Fusobacterium which does seem to be over represented in the severe group (group C).This does not completely invalidate the authors findings but serves to underscore that the sample size is small and different methods may give contrasting results based on the models employed. At least we can say there is some consistency as the Fusobacterium observation is reproduced. However, it is hard to know how reproducible or biologically important these findings are over all.

Reply 9: We very appreciate your comments. We did not find any difference in Lefse analysis also. The biological important of Fusobacterium in COPD need to further study to validate. Owing to the ethnic issues, we cannot put the clinical data to public domain. We are welcome to cooperate with other researcher with the some interest with us. We should provide our all original data for meta-analysis or further in-depth analysis.

Comment 10: I don’t think species level analysis is advisable in the case of 16S analysis. I would cut at genus level as it is more reliable and gives a more realistic appraisal of the microbiome in my view.

Reply 10: Thank you for your valuable comment. We modified the text in the results (pages 8-9) as you suggested. 

Comment 11: Though the taxonomic data is accessible, the authors stop short of making the clinical metadata available. Without this I can’t reproduce the bacteriodes finding. I would recommend the authors implement Maaslin (https://huttenhower.sph.harvard.edu/maaslin/) to see if the observations regarding lung function and eosinophil counts hold up. If they do, this is potentially an interesting finding.

Reply 11: Thank you for your valuable comments. We used spearman correlation coefficient to determine the relationship between each OTU abundance and clinical features. MaAsLin is a multivariate statistical framework that finds associations between clinical metadata and potentially high-dimensional experimental data. We are afraid small sample size is also a limitation to use multivariate statistical analysis.

Comment 12: Final line of the discussion goes too far. “This should be useful information for developing new diagnostic or therapeutic markers to control COPD progression.” The findings must be validated in independent studies with larger patient numbers before we are anywhere near even considering diagnostics or therapeutics. This is a very preliminary glimpse of what MIGHT be going on in COPD that requires extensive additional validation work in future studies.

Reply 12: Thank you for your kind comment. We modified the text in the discussion (pages 12) as you suggested.

---

## [Decision Letter · Decision Letter 1]

27 Jan 2021

PONE-D-20-30223R1

Comprehensive profiling of the gut microbiota in patients with chronic obstructive pulmonary disease of varying severity

PLOS ONE

Dear Dr. Wu,

Thank you for submitting your manuscript to PLOS ONE. After careful consideration, we feel that it has merit but does not fully meet PLOS ONE’s publication criteria as it currently stands. Therefore, we invite you to submit a revised version of the manuscript that addresses the points raised during the review process.

We look forward to receiving your revised manuscript.

Kind regards,

Aran Singanayagam

Academic Editor

PLOS ONE

Reviewers' comments:

Reviewer's Responses to Questions

**Comments to the Author**

1. If the authors have adequately addressed your comments raised in a previous round of review and you feel that this manuscript is now acceptable for publication, you may indicate that here to bypass the “Comments to the Author” section, enter your conflict of interest statement in the “Confidential to Editor” section, and submit your "Accept" recommendation.

Reviewer #1: All comments have been addressed

Reviewer #2: All comments have been addressed

2. Is the manuscript technically sound, and do the data support the conclusions?

Reviewer #1: Yes

Reviewer #2: Partly

3. Has the statistical analysis been performed appropriately and rigorously? 

Reviewer #1: Yes

Reviewer #2: Yes

4. Have the authors made all data underlying the findings in their manuscript fully available?

Reviewer #1: Yes

Reviewer #2: No

5. Is the manuscript presented in an intelligible fashion and written in standard English?

Reviewer #1: Yes

Reviewer #2: Yes

6. Review Comments to the Author

Reviewer #1: The Authors have clearly responded appropriately to the Reviewer comments, and the manuscript is strengthened through the changes made - thank you.

Reviewer #2: Major

Apologies, but I don’t see where the raw data has been deposited (e.g. NCBI sequence read archive). I think this is a PLoS requirement.

Minor

The authors state that “The R version 3.1.1 was used for heatmaps”.

That’s not really sufficient as one needs to know what package was used. The authors, at minimum, should specify what R packages were used in their analysis (e.g. pheatmap). A succinct summary (listing all R packages) in the methods would suffice.

Re: Reply 7:

If not already included somewhere, the authors should state the average DNA concentration observed in sample DNA extracts. If it is quite high (50ng/ul or more) this serves as a counterargument to the influence of contamination, which could also be highlighted in the discussion. While it is now field standard to control for contamination, and the exclusion of such samples is a major oversight, it would be comforting to know your DNA yields from stool were high, which presumably they were.

7. PLOS authors have the option to publish the peer review history of their article (what does this mean?). If published, this will include your full peer review and any attached files.

Reviewer #1: **Yes: **Benjamin H Mullish

Reviewer #2: No

---

## [Author Response · Author response to Decision Letter 1]

2 Feb 2021

Reviewer #2: Major

Apologies, but I don’t see where the raw data has been deposited (e.g. NCBI sequence read archive). I think this is a PLoS requirement.

Response: In this study, we reported the analyses of 16S rRNA V3+V4 hypervariable regions by sequencing for stool from COPD patients. The data is neither microarray nor deep sequencing data. We did not found any public database which enrolled microbiota profile using partial sequence (V3+V4) data.

Data Availability: The data has been also submitted as S1 Table (Excel file) in Supporting Information files. The data file contains the reads number blast to the OTUs in each sample. The sample ID: A- x = samples in group A; B- x = samples in group B; C- x = samples in group C

Some articles had been published in Plos One (as following) did not mention any deposited (e.g. NCBI sequence read archive) record.

Ingrid S. Surono, Dian Widiyanti, Pratiwi D. Kusumo, Koen Venema. Gut microbiota profile of Indonesian stunted children and children with normal nutritional status. Published: January 26, 2021https://doi.org/10.1371/journal.pone.0245399

Aasia Khaliq, Resmi Ravindran, Samia Afzal, Prasant Kumar Jena, Muhammad Waheed Akhtar, Atiqa Ambreen, Yu-Jui Yvonne Wan, Kauser Abdulla Malik, Muhammad Irfan, Imran H. Khan Gut microbiome dysbiosis and correlation with blood biomarkers in active-tuberculosis in endemic setting. Research Article | published 22 Jan 2021 PLOS ONE https://doi.org/10.1371/journal.pone.0245534

We are willing to share our data by fasta format under a reasonable requirement in the future. 

Minor

The authors state that “The R version 3.1.1 was used for heatmaps”.

That’s not really sufficient as one needs to know what package was used. The authors, at minimum, should specify what R packages were used in their analysis (e.g. pheatmap). A succinct summary (listing all R packages) in the methods would suffice.

Response:

Thank you for the suggestion. The statement has been revised as “The pheatmap package (https://cran.r-project.org/src/contrib/Archive/pheatmap/) was used for ecological analysis and heatmaps”. (Page 7)

Re: Reply 7:

If not already included somewhere, the authors should state the average DNA concentration observed in sample DNA extracts. If it is quite high (50ng/ul or more) this serves as a counterargument to the influence of contamination, which could also be highlighted in the discussion. While it is now field standard to control for contamination, and the exclusion of such samples is a major oversight, it would be comforting to know your DNA yields from stool were high, which presumably they were.

Response:

We added the DNA extraction kit in methods section (page 4). The average DNA concentration has been stated in results section as “The stool DNA samples were eluted in 200 μl AE buffer. The average of DNA concentration was 5.52 ng/ul (range 1.05 – 12.43 ng/ul).” Please see page 8.

---

## [Decision Letter · Decision Letter 2]

22 Feb 2021

PONE-D-20-30223R2

Comprehensive profiling of the gut microbiota in patients with chronic obstructive pulmonary disease of varying severity

PLOS ONE

Dear Dr. Wu,

Thank you for submitting your manuscript to PLOS ONE. After careful consideration, we feel that it has merit but does not fully meet PLOS ONE’s publication criteria as it currently stands. Therefore, we invite you to submit a revised version of the manuscript that addresses the points raised during the review process.

ACADEMIC EDITOR: The manuscript is almost ready for acceptance but the comment made by reviewer 2 needs to be addressed. It is now field standard for all 16S rRNA sequencing data to be uploaded onto a public repository server (eg European Nucleotide Archive) so that other researchers have open access to the data. Please organise for your data to be submitted and available in this way.

We look forward to receiving your revised manuscript.

Kind regards,

Aran Singanayagam

Academic Editor

PLOS ONE

Reviewers' comments:

Reviewer's Responses to Questions

**Comments to the Author**

1. If the authors have adequately addressed your comments raised in a previous round of review and you feel that this manuscript is now acceptable for publication, you may indicate that here to bypass the “Comments to the Author” section, enter your conflict of interest statement in the “Confidential to Editor” section, and submit your "Accept" recommendation.

Reviewer #2: (No Response)

2. Is the manuscript technically sound, and do the data support the conclusions?

Reviewer #2: Partly

3. Has the statistical analysis been performed appropriately and rigorously? 

Reviewer #2: Yes

4. Have the authors made all data underlying the findings in their manuscript fully available?

Reviewer #2: No

5. Is the manuscript presented in an intelligible fashion and written in standard English?

Reviewer #2: Yes

6. Review Comments to the Author

Reviewer #2: "We did not found any public database which enrolled microbiota profile using partial sequence (V3+V4) data."

I cannot see any reason that public repositories such as the sequence read archives (NCBI) would not accept the type of data you describe. I highly encourage the authors to make their data publicly available as per field standards.

7. PLOS authors have the option to publish the peer review history of their article (what does this mean?). If published, this will include your full peer review and any attached files.

Reviewer #2: No

---

## [Author Response · Author response to Decision Letter 2]

16 Mar 2021

Reviewer #2: 

I cannot see any reason that public repositories such as the sequence read archives (NCBI) would not accept the type of data you describe. I highly encourage the authors to make their data publicly available as per field standards.

ACADEMIC EDITOR: The manuscript is almost ready for acceptance but the comment made by reviewer 2 needs to be addressed. It is now field standard for all 16S rRNA sequencing data to be uploaded onto a public repository server (eg European Nucleotide Archive) so that other researchers have open access to the data. Please organise for your data to be submitted and available in this way.

Response:

We have organized our data and submitted to European Nucleotide Archive, Accession No: PRJEB43280. Please see the section “Availability of data and materials”.

---

## [Editor Report · Decision Letter 3]

29 Mar 2021

Comprehensive profiling of the gut microbiota in patients with chronic obstructive pulmonary disease of varying severity

PONE-D-20-30223R3

Dear Dr. Wu,

We’re pleased to inform you that your manuscript has been judged scientifically suitable for publication and will be formally accepted for publication once it meets all outstanding technical requirements.

Kind regards,

Aran Singanayagam

Academic Editor

PLOS ONE
---

## [Editor Report · Acceptance letter]

31 Mar 2021

PONE-D-20-30223R3 

Comprehensive profiling of the gut microbiota in patients with chronic obstructive pulmonary disease of varying severity 

Dear Dr. Wu:

I'm pleased to inform you that your manuscript has been deemed suitable for publication in PLOS ONE. Congratulations! Your manuscript is now with our production department. 

Kind regards, 

on behalf of

Dr. Aran Singanayagam 

Academic Editor

PLOS ONE